# Exercise May Affect Metabolism in Cancer-Related Cognitive Impairment

**DOI:** 10.3390/metabo10090377

**Published:** 2020-09-20

**Authors:** Muhammad Shahid, Jayoung Kim

**Affiliations:** 1Departments of Surgery and Biomedical Sciences, Cedars-Sinai Medical Center, Davis 5071, 8700 Beverly Blvd., Los Angeles, CA 90048, USA; muhammad.shahid@cshs.org; 2Samuel Oschin Comprehensive Cancer Institute, Cedars-Sinai Medical Center, Los Angeles, CA 90048, USA; 3Department of Medicine, University of California Los Angeles, Los Angeles, CA 90024, USA; 4Department of Urology, Ga Cheon University College of Medicine, Incheon 461-701, Korea

**Keywords:** cancer-related cognitive impairment, therapeutic exercise intervention, metabolism, metabolomics profiling

## Abstract

Cancer-related cognitive impairment (CRCI) is a significant comorbidity for cancer patients and survivors. Physical activity (PA) has been found to be a strong gene modulator that can induce structural and functional changes in the brain. PA and exercise reduce the risk of cancer development and progression and has been shown to help in overcoming post-treatment syndromes. Exercise plays a role in controlling cancer progression through direct effects on cancer metabolism. In this review, we highlight several priorities for improving studies on CRCI in patients and its underlying potential metabolic mechanisms.

## 1. Introduction

From the total global population, 13% are adults 60 years or older, which accounts for approximately 962 million people. This group is predicted to steadily increase in population to 1.4 billion, 2.1 billion, and 3.1 billion by 2030, 2050, and 2100, respectively [1]. Patients with cognitive impairment (CI) make up a large number of adults in the 60 years or older population. These patients may experience difficulties in daily functioning, decision-making, and treatment adherence; thereby, leading to reduced quality of life (QoL) and decreased survival [2,3]. In an effort to maintain independence in older adults, focusing on cognitive function is a novel target of concern, since the origins of cognitive decline may be reversible or even treatable. As a result, understanding cognitive decline in older adults is a growing field of interest. CI can also lead to increases in caregiver burdens. Prevention of CI in cancer patients is especially important for older patients, since there have been notable increases in long-term survival due to new treatments and a resulting growing number of people living with cancer as a chronic condition. The spectrum of cognitive decline between normal cognitive and mild cognitive impairment (MCI) to dementia in older adults can range from natural cognitive decline due to age to atypical cognitive impairment [1]. The prevalence rate in the group of adults older than 60 MCI showed increasing with age and lower levels of education which is approximately 6.7% to 25.2%, and is more prevalent in men [4,5].

## 2. CI in Cancer Patients

Cancer and treatments for it, including chemotherapy, hormone therapy, and radiation therapy, can have harmful effects on mental processes [6,7]. Previous studies reported a higher incidence of cognitive dysfunction among cancer patients compared to healthy matched controls. Up to 85% of cancer patients receiving treatment have been found to report mild to severe cognitive complaints, which can last months to even years after finishing treatment [8]. Cancer-related CI (CRCI) can be classified as subtle, moderate, or severe based on neuropsychological testing. CRCI is the most frequent complication reported by breast cancer patients [9]. Cognitive complaints have been reported by more than 50% of breast cancer patients following chemotherapy; however, only 15%–25% of these patients have shown objective cognitive decline [10]. Demographic and other health factors, such as age, race, socioeconomic status, education, menopausal status, and body mass index, are also known to affect cognition in adults [11,12]. Since difficulties in cognitive function have a negative impact on QoL (autonomy, work balance, relationships, and self-image), there is an urgent need for more pronounced CRCI management in patients. This has fueled studies on potentially implementing cognitive rehabilitation for cancer patients [13]. In light of the prevalence and associated individual burden of CRCI, there is a clear need for strategies to manage CRCI. Currently, there are no established treatment options to reduce CRCI risk or diminish its severity [14,15]. Furthermore, advancements in hormone therapy, targeted therapy, and immunotherapy have resulted in greater survival rates for cancer patients, but at the cost of increased potential cognitive impacts [16]. 

The precise mechanisms underlying the pathophysiology of CRCI are unclear. Demographic factors, including age, race, socioeconomic status, and education, as well as menopausal status, health status, and body mass index, are also known to affect cognition in adults. In light of the prevalence and associated individual burden of CRCI, there is a clear need for strategies to manage CRCI. Currently, no established treatment options exist to reduce the risk of CRCI or diminish its severity. Campbell et al., performed a systematic analysis of 29 randomized controlled trials (RCTs) to better understand the relationship of exercise with CRCI. In 12 of these trials (41%) (Cohen d range: 0.24–1.14), they found that exercise had a significant effect on self-reported cognitive function during and after chemotherapy. These 12 trials used the EORTC QLQ-C30 exam for cognitive functioning. In 10 other trials (34%), neuropsychological testing was used to evaluate cognitive functioning; however, only 3 of these trials in breast cancer reported significant benefits from exercise (Cohen d range: 0.41–1.47) [17]. 

Another study by Witlox et al. found that physical exercise had positive effects in healthy older adults and those with mild cognitive impairment. They recruited 180 breast cancer patients with cognitive issues 2–4 years after their diagnosis with cancer and randomized them (1:1) into two groups; exercise intervention and control. The exercise intervention group underwent a 6-month course of twice weekly 1 h supervised aerobic and strength-training exercises with twice weekly 1 h power walking. They concluded that physical exercise improves cognitive functioning for breast cancer survivors [18].

To better understand CRCI’s pathophysiology and the direct impact of different cancer treatments, animal models have been developed [19].

## 3. Biological Drivers of CRCI

Evidence from clinical and pre-clinical research suggests that many mechanisms play a role in the development of CRCI, including inflammation [20,21]. Inflammation is an important mechanism underlying cognitive impairment, especially in the elderly. Accumulating evidence has linked inflammation to cognitive decline and the risk of dementia [22]. Chemotherapy-induced pro-inflammatory cytokines levels of interleukin (IL)-1β, IL-6, tumor necrosis factor-α (TNF-α), and IL-10 in cancer patients have been related to possible CI [23,24]. A large multicentered cohort study conducted in Singapore found that among proinflammatory plasma cytokines—including IL-1β, IL-2, IL-4, IL-6, IL-8, IL-10, granulocyte-macrophage colony-stimulating factor, interferon-γ, and TNF-α—elevated IL-1β and IL-6 were associated with greater self-reported cognitive impairments (*p* = 0.018 and 0.001, respectively) [25]. However, the effects of cytokines in post-chemotherapy cognitive impairment remains in controversy; other studies have published conflicting results about the relationship between cytokine concentration and cognition [26,27]. Nonsteroidal anti-inflammatory drugs (NSAIDs) have often been administered as a preventative measure against Alzheimer’s disease (AD). The mechanism of action is believed to be through blockage of cyclo-oxygenase isoforms (e.g., COX-1 and COX-2). NSAIDs have neurotoxic and neuroprotective effects with diverse impacts on mechanisms that may influence cognitive impairment, including inflammation, release of neurotransmitters, synaptic plasticity, cerebral ischemia, and functioning of cerebral endothelial and smooth muscle [28]. 

Oxidative stress and its associated damage in the age-dependent cognitive loss has been previously highlighted [29]. Chemotherapy-induced oxidative stress-mediated TNF-α triggers inducible nitric oxide synthase (iNOS) production [30]. Apolipoprotein A-I (ApoA1) is possibly one of the key factors in oxidative stress and pro-inflammatory cytokine mediated CRCI. Oxidation and down-regulated expression of ApoA1 were found in a number of neurodegenerative diseases with cognitive deficits, such as Alzheimer’s and Parkinson’s Diseases [31]. Administration of vitamin E ameliorated memory deficit. Vitamin E-deficient rats showed decreased learning as well as memory retention ability, whereas younger rats supplemented with vitamin E displayed accelerated learning and capabilities. This could be that learning ability declined gradually with age due to chronic exposure to oxidative stress [32]. The association of DNA damage with aging is well-studied [33]. DNA lesions are accumulated in the brain during AD. Elevations in γH2AX, a well-established marker of double-strand breaks (DSB) [34], were detected in 11 of 13 AD brains in the astrocytes of the hippocampus and cerebral cortex [35]. Studies on two independent cohorts (*n* = 13 and *n* = 23) found significant increases of γH2AX in the astrocytes and neurons of the hippocampus and frontal cortex of AD brains; this increase was found in brains with MCI as well [36]. Other factors include reduced synaptic plasticity, altered growth factor levels, and impaired hippocampal neurogenesis [37,38,39].

## 4. The Interrelationship between CRCI and Alterations in Metabolism

In addition to powering live systems, metabolism is a complex phenomenon that is tightly linked to signaling pathways, post-translational modifications, and gene expression. In general, metabolism acts as a cellular rheostat [40]. Metabolism supports a variety of normal cell functions, including breakdown of carbohydrates, fats, and amino acids to generate energy and biosynthetic precursors needed for growth [41]. The fundamental features of metabolism are reprogrammed in cancer cells to support their aberrant growth and proliferation. This change in functioning is likely the result of genomic alterations (i.e., mutations in oncogenes and tumor suppressor genes), tumor microenvironment (compromised nutrients and oxygen availability), and other factors [42]. Metabolic alterations are a hallmark of cancer [43]. These changes favor rapidly dividing cells, inhibit the prevention of tumor initiation, and attenuates proliferation and metastasis [44]. In order to better understand cancer-specific metabolism, a systemic application of analytical techniques that can assess metabolic levels is needed. 

Metabolomics combines high-throughput analysis with bioinformatics and aims to comprehensively analyze all metabolites in a given biological sample. Over the past 20 years, incredible advancements have been made in metabolomics. This has propelled its evolution into becoming a powerful tool in medicine and science, especially in the study of disease-related biomarkers, toxicology, and molecular mechanisms. Metabolomics can also provide greater detailed information regarding human biochemistry [45,46]. Metabolomics is commonly applied to discover diagnostic, prognostic, or therapeutic biomarkers [47]. For instance, early metabolomic experiments in breast cancer patients lead to the identification of positive associations between choline, glycine, and lactate with tumor grade and size [48]. Similar work has now been done in ovarian [49], prostate [50], and various other cancers. Since the efforts of metabolomics-based biomarker discovery have been well summarized in previously published review papers as described above, we will not further discuss the topic in this article. 

Since its first observation 90 years ago by Dr. Otto Warburg, one of the hallmarks in cancer biology is the metabolic reprogramming such as high rates of aerobic glycolysis [51,52,53,54]. Cancer-associated metabolic alteration has been reported in many types of cancers. Amino acids, such as arginine, proline, glutamine, and creatine, play important roles as substrates and protein synthesis in cancer cells and they have been widely studied in cancer metabolism. Specific metabolic pathways altered by cancer have also been reported, which provide therapeutical strategies. Metabolic reprogramming is considered for regimens an exploited for cancer therapy. Amino acid depletion therapies are being tested against several cancer types [55,56]. For example, metformin, an inhibitory drug of mitochondrial metabolism, has been suggested to have a synergistic effect when used with chemotherapies by inhibiting proliferation of cancer cells [57,58].

Doxorubicin (DOX) is a widely used antitumor agent for the treatment of a series of cancer types. However, DOX shows cytotoxicity to noncancer cells such as heart, skeletal muscle, liver, and kidney cells, leading to adverse effect. Recently, exercise is reported to show a beneficial adaptation to reduce the DOX-induced cellular toxicity [59]. However, the mechanisms underlying the exercise-induced protection against DOX cytotoxicity are not clear. Exercise has another benefit, reducing cognitive impairment. Brain-derived neurotrophic factor (BDNF), a key mediator of cognitive impairment in Alzheimer’s dementia. A recent finding suggested that exercise-induced expression of BDNF, suggesting the potential mechanism of exercise benefit to increase cognitive function [60]. 

Prior studies have also provided essential information on the occurrence and association of metabolic alterations and cognitive impairment in humans. However, they have not established the molecular mechanisms behind these relationships, nor the therapeutic window that would allow for treatment before irreversible damage occurs in both systemic functionality and cognitive skills. 

Different lifestyle factors are involved in the arise of diseases which may also lead increasing the risk of developing AD. Lifestyle factors are increasingly being recognized for their role in figuring out cognitive impairment, or the lack thereof, with age. Participation in recreational physical activity (PA) has been shown to be inversely related to the prevalence of age-related cognitive decline and dementia [61]. Independent studies have repeatedly shown that PA reduces cognitive decline in older age groups [62]. Similarly, several studies have also shown that due to the limited use of diet or diet restraint is resulted with the maintenance of cognitive function later in life [63]. 

Omega-3 fatty acids and antioxidants are specific dietary components which preserves cognitive function. Daily intake of these components works well for the eighth decade of life [64]. Glucose and insulin metabolism are key metabolic pathways and represents a metabolic spectrum with clinical thresholds for prediabetes and diabetes mellitus. Further increases in fasting glucose, with concomitant impairment of both glucose and insulin metabolism, are hallmarks of type 2 diabetes. Obesity is a known significant risk factor of AD progression for both prediabetes and diabetes [63]. Possible mechanisms underlying metabolic reserves in MCI and AD involve direct roles for insulin [65], insulin-like growth factors [66], and neurotrophic factors [67], as well as pathological changes in glucose metabolism and protein glycosylation [68]. Other components of the hormonal milieu include adipocyte cytokine leptin [69].

## 5. Role of Exercise in CRCI 

According to the World Health Organization’s definition, any bodily movement that involved skeletal muscles by utilizing energy is called PA; whereas, well planned, structured, repetitive, and intentional movement known as exercise, which is a subcategory of PA. Most observational studies assess PA rather than exercise. Based on multiple meta-analyses, physical exercise is known to be crucial for maintaining general health [70,71]. Exercise helps maintain body weight and reduces stress. People who regularly exercise are less likely to smoke tobacco or overeat. Moreover, exercise directly targets the primary aspects of health, including heart function, cholesterol, triglycerides, blood pressure, and brain function. It has been shown that regular exercise leads to enhanced maximal oxygen uptake and increases the mean lifespan of laboratory animals and humans [72]. Exercise is known to impact almost every system in the body. Benefits include improved cardiovascular health, greater bone mineral density (BMD), and decreased risk of cancer, stroke, diabetes, and cognitive impairment. 

Exercise is an established safe and effective therapy for managing numerous adverse effects of cancer treatment, including fatigue, psychological distress, functional decline, and detrimental body composition changes [73]. Accumulating evidence on the positive role of exercise on improving cognitive function in healthy older adults and those with mild cognitive impairment or more severe neurocognitive impairment (i.e., AD, stroke) has sparked significant interest in the potential use of exercise as an effective management strategy for CRCI [74,75,76]. A variety of evidence supports the conclusion that exercise link to cancer which could lower the risk of different cancers including colon, breast, kidney, endometrial, bladder, esophageal, and stomach cancers, with moderate evidence for lung cancer. The Physical Activity Guidelines Advisory Committee determined and vigorously appreciated the conclusion [77]. 

Unfortunately, due to limited preclinical studies and testing of the antitumor activity of exercise is restricted. So far, only 53 studies were reported in vivo preclinical testing to assess the activity of various exercise paradigms on tumor incidence, growth, or metastasis; 35 of these studies positively reported that exercise inhibited cancer growth or progression [78]. More recently, higher-quality studies have demonstrated that paradigms of PA showed a link between exercise and epinephrine, and IL-6 to NK cell mobilization and redistribution; both of which ultimately control tumor growth [79]. 

Benefits of exercise were particularly noted on self-reported cognitive function in women with breast cancer [80]. RCTs have suggested that people should adopt PA and exercise to alleviate the negative impacts of aging on cognitive function. Through a meta-analysis, Heyn P. et al. found that PA and exercise had positive effects on cognition among those with cognitive decline [80]. 

## 6. The Effects of Exercise on Cancer Metabolism and Its Associated Signaling Pathways

The role of PA and exercise in the whole process of cancer from prevention to post-treatment has been extensively studied [70,81]. There is ample evidence suggesting regular PA to be related to a reduced risk for various forms of cancer [82]. Exercising at varying intensities has been found to have remarkable effects on physiologic and gene expression adaptations in mammals [83]. The role of exercise has surprisingly received little to no attention in high-risk individuals. However, recently, a larger number of groups are investigating the effects of PA and exercise on cancer; a field now called “exercise-oncology” [84]. In cancer patients, exercise is now well-documented to be a tolerable adjunct therapy associated with significant benefits across a wide range of symptoms [85]. 

Intratumoral signaling networks are highly modifiable and modulated by numerous extrinsic factors [86]. Vulczak et al. demonstrated that the mitochondrial activity of tumor cells in animals that exercised was lower in comparison to the tumor cells of sedentary animals, with a significant decrease in the electron transport chain capacity (E). This demonstrates lower respiratory capacity independent of mitochondrial content, measured by citrate synthase (CS) activity [87]. Kynurenine (KYN), a catabolite of the amino acid tryptophan (TRP), was found to be associated with progression and poor clinical outcome in numerous cancer types [88]. Zimmer et al. and his team investigated the influence of resistance exercise on the KYN pathway in breast cancer patients. They showed the potential exercise-induced modulation of KYN pathway metabolites in the serum and urine of healthy women and breast cancer patients undergoing radiotherapy [89]. The Akt/mTOR pathway is central for controlling growth and protein synthesis and plays a pivotal role in the muscular response to resistance training [90]. Thompson et al. reviewed several preclinical studies highlighting how the Akt/mTOR pathway is differentially regulated with exercise in many tumor types [91]. 

## 7. Conclusions

In conclusion, current evidence suggests that physical exercise shows much promise in improving cognitive impairment among cancer patients and survivors. Exercise also shows much influence on cancer incidence, lowers the risk of recurrence, and secures longer higher quality life for patients. Most of cancer patients expressed their preferences in the therapeutic potential of exercise over chemotherapeutics; this strategy may potentially alter disease pathogenesis and symptoms without the adverse effects of conventional pharmacological agents. The positive effects of exercise are evident in large epidemiological studies, as well as controlled laboratory studies. Moreover, exercise inhibits tumor growth across cancers and at all stages of tumor development. Given these findings, future research needs to consider the type of measurements used to measure CRCI, which will further improve patient care and lead to the development of targeted therapies, preventative strategies, and cognitive rehabilitation treatment.

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
