# Peer review of "Exercise May Affect Metabolism in Cancer-Related Cognitive Impairment"

_metabolites, 2020, doi:10.3390/metabo10090377_

Round 1

Reviewer 1 Report

As a review it reads well and covers the main areas of interests within the subject discussed. The work provides some thought on where the topic should be heading in the future and is a good summary of where we are at at this time within the topic. It would be good to see this translated into a systematic review with analysis of the data. 

Lines 221 and 223 have superscript numbers 60 and 61 I suspect these are supposed to be references from a previous version and the Harvard style reference needs to be substituted.

Line 146 and 147 suggests work have previously been shown in other reviews it would be nice if some of these were highlighted for the general reader to follow up.

Author Response

As a review, it reads well and covers the main areas of interest within the subject discussed. The work provides some thought on where the topic should be heading in the future and is a good summary of where we are at this time within the topic.

[Response] Thank you for this positive comment.

  1. Lines 221 and 223 have superscript numbers 60 and 61 I suspect these are supposed to be references from a previous version and the Harvard style reference needs to be substituted.

[Response] We apologize for this mistake in our previous manuscript. We have revised the manuscript accordingly (page 7).

  1. Line 146 and 147 suggests work have previously been shown in other reviews it would be nice if some of these were highlighted for the general reader to follow up.

[Response] Thanks for reviewer 1’s constructive comments. Based on the reviewer’s comment, we revised the manuscript accordingly and have added information from the aforementioned articles (pages 4- 5).

Reviewer 2 Report

This study is weak. First of all, there is no clear or strong evidence that suggests cognitive impairment is directly related to cancer. 

The authors mentioned that inflammation and oxidative stress might be the biological drivers of Cancer-related cognitive impairment. But again, there is no solid evidence to back up this claim. 

Last but not least, the authors also mention that exercise is beneficial to cancer treatments and reduced cancer risk. But it is difficult to tell whether it is a direct effect or an indirect effect via reducing cognitive impairment.

Overall, this work does not provide much new knowledge. The logical flow is weak and I feel I am not convinced that exercise can play in important role in Cancer‑Related Cognitive Impairment.

Author Response

  1. The authors mentioned that inflammation and oxidative stress might be the biological drivers of Cancer-related cognitive impairment. But again, there is no solid evidence to back up this claim.

[Response] Thank you for bringing up this important point. we greatly appreciate the reviewer’s positive comments on our study. We added more reference papers suggesting the inflammatory and oxidative stress linked to the CRCI. We revised the manuscript as recommended and included discussions according to the reviewer’s comment (pages 4-5).

  1. Last but not least, the authors also mention that exercise is beneficial to cancer treatments and reduced cancer risk. But it is difficult to tell whether it is a direct effect or an indirect effect via reducing cognitive impairment.

[Response] If our previous manuscript was not able to deliver the message clearly enough, we apologize for that. There are many examples suggesting exercise is beneficial to cancer treatments and reduced cancer risk by reducing cognitive impairment.  We heavily revised our manuscript accordingly (Page 7).

  1. Overall, this work does not provide much new knowledge. The logical flow is weak and I feel I am not convinced that exercise can play an important role in Cancer‑Related Cognitive Impairment.

[Response] This is a great comment. To improve our argument, we have revised the manuscript to improve the clarity and flow of the text and included discussions on recent publications as the reviewer recommended (page 4).

Round 2

Reviewer 2 Report

The authors well addressed my concerns. I do not have further comment.

Author Response

Thank you.